A retrospective approach for evaluating ecological niche modeling transferability over time: the case of Mexican endemic rodents

http://orcid.org/0000-0002-4462-3176 Moreno-Arzate Claudia N. 1 2
Martínez-Meyer Enrique 3 4 emm@ib.unam.mx
1 Instituto de Ecología, Universidad Nacional Autónoma de México , Mexico City , Mexico
2 Posgrado en Ciencias Biológicas, Universidad Nacional Autónoma de México , Mexico City , Mexico
3 Departamento de Zoología, Instituto de Biología, Universidad Nacional Autónoma de México , Mexico City , Mexico
4 Laboratorio Nacional Conahcyt sobre la Biología del Cambio Climático , Mexico City , Mexico
Roper James
Electronic publication date: 2024 Nov 29
Publication date: 2024
Volume: 12
Electronic Location ID: e18414
Received 2024 Apr 28; Accepted 2024 Oct 7
Copyright: © 2024 Moreno-Arzate and Martínez-Meyer
Copyright year: 2024
Copyright holder: Moreno-Arzate and Martínez-Meyer
License: This is an open access article distributed under the terms of the Creative Commons Attribution License, which permits unrestricted use, distribution, reproduction and adaptation in any medium and for any purpose provided that it is properly attributed. For attribution, the original author(s), title, publication source (PeerJ) and either DOI or URL of the article must be cited.
License URL: https://creativecommons.org/licenses/by/4.0/

Keywords: Ecological niche modeling, Climate change, Modeling algorithms, Mammals, Mexico, Endemic rodents, Model transferability

Funding: Consejo Nacional de Humanidades, Ciencias y Tecnologías of Mexico (CONAHCYT) 211508 Claudia N. Moreno-Arzate received scholarship no. 211508 from the Consejo Nacional de Humanidades, Ciencias y Tecnologías of Mexico (CONAHCYT). The funders had no role in study design, data collection and analysis, decision to publish, or preparation of the manuscript.

==============================
Ecological niche modeling (ENM) is a valuable tool for inferring suitable environmental conditions and estimating species’ geographic distributions. ENM is widely used to assess the potential effects of climate change on species distributions; however, the choice of modeling algorithm introduces substantial uncertainty, especially since future projections cannot be properly validated. In this study, we evaluated the performance of seven popular modeling algorithms—Bioclim, generalized additive models (GAM), generalized linear models (GLM), boosted regression trees (BRT), Maxent, random forest (RF), and support vector machine (SVM)—in transferring ENM across time, using Mexican endemic rodents as a model system. We used a retrospective approach, transferring models from the near past (1950–1979) to more recent conditions (1980–2009) and vice versa, to evaluate their performance in both forecasting and hindcasting. Consistent with previous studies, our results highlight that input data quality and algorithm choice significantly impact model accuracy, but most importantly, we found that algorithm performance varied between forecasting and hindcasting. While no single algorithm outperformed the others in both temporal directions, RF generally showed better performance for forecasting, while Maxent performed better in hindcasting, though it was more sensitive to small sample sizes. Bioclim consistently showed the lowest performance. These findings underscore that not all species or algorithms are suited for temporal projections. Therefore, we strongly recommend conducting a thorough evaluation of the data quality—in terms of quantity and potential biases—of the species of interest. Based on this assessment, appropriate algorithm(s) should be carefully selected and rigorously tested before proceeding with temporal transfers.

Introduction

Climate change is significantly impacting biodiversity, causing shifts in species abundance and distribution that lead to extensive reshuffling of biotas (Parmesan & Yohe, 2003; Pacifici et al., 2015; Peterson et al., 2015; MacLean et al., 2018; Widick & Bean, 2019; Habibullah et al., 2022; Jaroszynska et al., 2023). In response to these dynamic changes, ecological niche modeling (ENM) has emerged as a valuable tool for analyzing and predicting species’ geographic distributions under various climatic scenarios, both past and future. Grounded in ecological niche theory, ENM integrates methodologies from multiple disciplines, including informatics, geographic information systems (GIS), and statistics (Austin, 2007; Soberón & Nakamura, 2009; Wiens et al., 2009; Sillero et al., 2021). This correlative approach utilizes georeferenced species occurrence data and environmental predictor variables to model the ecological niche of species, and projecting this information onto geographic space to generate a map that is commonly referred to as the species’ potential distribution (Barbet-Massin & Jetz, 2014; Elith et al., 2006; Peterson, 2011; Soley-Guardia, Alvarado-Serrano & Anderson, 2024).

Several methods for constructing niche models have evolved to address different types of occurrence data: presence-only (e.g., Bioclim, environmental distances), presence-absence (e.g., generalized linear models (GLM), generalized additive models (GAM)), presence-pseudoabsence (e.g., Genetic Algorithm for Rule-set Prediction (GARP)), and presence-background (e.g., Ecological Niche Factor Analysis (ENFA), Maxent) (Phillips, 2008; Barbet-Massin et al., 2012; Barbet-Massin & Jetz, 2014; Warton & Aarts, 2013; Fan et al., 2018; Qiao et al., 2019; Sillero et al., 2023). Presence-absence methods are considered more robust when unequivocal absence data are available, as they help identify unsuitable areas that might be misclassified by presence-only methods (Brotons et al., 2004; Golicher et al., 2012). However, reliable absence data are rare, leading to the predominance of presence-only, presence-pseudoabsence, and presence-background algorithms (Vaz, Cunha & Nabout, 2015; Soley-Guardia, Alvarado-Serrano & Anderson, 2024; Sillero et al., 2021).

ENM is extensively used to evaluate the potential impacts of climatic changes by projecting modeled niches under current conditions onto past or future scenarios, a concept known as model transferability (Pearson & Dawson, 2003; Peterson, Martínez-Meyer & González-Salazar, 2004; Thomas et al., 2004; Waltari & Guralnick, 2009; Heikkinen, Marmion & Luoto, 2012; Garcia et al., 2016; Zhu, Fan & Peterson, 2021; Sillero et al., 2021). The success of this transferability lies on three critical assumptions: (1) the ecological niche remains stable during the period of transfer (Soberón & Nakamura, 2009); (2) the relationship between environmental variables and species remains constant during climatic changes (Hijmans & Graham, 2006; Wiens et al., 2009); and (3) the species is in equilibrium with the environment in the calibration scenario, meaning they occupy the available suitable areas accessible to them (Sequeira et al., 2018; Yates et al., 2018; Werkowska et al., 2017). These premises are critical as they assume that the current relationship between the species and the environment is optimal, and transferability will be defined by how closely the model conforms to this relationship.

A major challenge in model transferability is the presence of non-analog climates—environmental conditions in the projected climatic scenario that are absent in the calibration scenario (Sequeira et al., 2018). Algorithms respond idiosyncratically to these conditions due to their programmatic features, frequently producing disparate results (Pearson, 2006; Araújo & Peterson, 2012; Essl et al., 2023). This issue is particularly problematic when projecting ecological niches into future climates, where no empirical data are available to validate algorithm performance. Consequently, a common approach for evaluating algorithm effectiveness in climate change studies involves projecting models between two historical periods for which data are available, such as from the recent past to the present (Rubidge et al., 2011; Piirainen et al., 2023).

Extensive research has been conducted to compare algorithm performance across spatial and temporal transferences, utilizing both real and virtual species (Prasad, Iverson & Liaw, 2006; Hijmans & Graham, 2006; Kharouba, Algar & Kerr, 2009; Dobrowski et al., 2011; Rubidge et al., 2011; Moreno-Amat et al., 2015; García-Callejas & Araújo, 2016). These studies reveal that results can vary greatly due to factors such as species traits, biotic interactions, data completeness, and climatic dissimilarities. Such variations complicate the identification of the specific impacts that algorithm choices have on the outcomes (Yates et al., 2018; Merow et al., 2014).

In this study, we evaluated the performance of seven popular modeling algorithms—Bioclim, GAM, GLM, boosted regression trees (BRT), Maxent, random forest (RF), and support vector machine (SVM)—in transferring niche models of Mexican endemic rodents from the mid-20th century to the late-20th/early 21st centuries and vice versa. We selected Mexican endemic rodents due to their restricted and relatively well-known distributions, and because we do not expect significant climatic niche evolution over this period (Martínez-Meyer, Peterson & Hargrove, 2004). We expected that species with poor data quality (e.g., scarce or biased) would produce poor results regardless of the algorithm used. Conversely, for well-sampled species, we hypothesized that algorithms capable of modeling biologically meaningful response curves (e.g., bell-shaped responses to temperature), such as Maxent or SVM, would outperform simpler algorithms like Bioclim or GLM.

Materials and Methods

Occurrence data of species

We compiled occurrence records for 117 Mexican endemic rodent species (Ramírez-Pulido et al., 2014; Supplemental Material, Table S1) from various sources including natural history collections, journal articles, books, and theses (Supplemental Material, Table S2). Records span two periods, 1950–1979 and 1980–2009, to align with available climatologies for Mexico. We eliminated duplicate records and those with questionable taxonomic or geographic certainty, retaining species with at least 10 unique localities per period to reduce the risk of data incompleteness issues (Hernandez et al., 2006). For algorithms requiring absence data, we generated pseudoabsences by randomly selecting non-presence localities matching the number of presence records using the Ecospat package in R 3.5, (Di Cola et al., 2017; R Core Team, 2017).

Climatic variables

We used 19 bioclimatic variables generated for Mexico for the mid-20th century (Time 1 (T1): 1950–1979) and the late 20th/early 21st centuries (Time 2 (T2): 1980–2009) (Cuervo-Robayo et al., 2020). These variables were derived from monthly averages of precipitation and minimum and maximum temperatures recorded at climate stations across Mexico, southern USA, northern Guatemala, and Belize. The resulting surfaces have a spatial resolution of 30 arc seconds (~1 km), following the methodology of the WorldClim dataset (Hijmans et al., 2005), and summarize the extreme, mean, and seasonal patterns in temperature and rainfall. To reduce model complexity and minimize overfitting, we conducted Pearson correlation analyses for each species and excluded variables with correlations above 0.80 (Radosavljevic & Anderson, 2014; Moreno-Amat et al., 2015; García-Callejas & Araújo, 2016; Regos et al., 2019) (Supplemental Material, Table S3). The area of analysis for each species resembles its accesibility (i.e., “M” in the BAM framework; Barve et al., 2011), and it was delineated by clipping the raster layers to the ecoregions (Olson et al., 2001) where each species has been recorded, assuming ecoregion boundaries serve as dispersal barriers (Radosavljevic & Anderson, 2014).

Ecological niche modeling

We evaluated the transfer capacity of seven algorithms: Bioclim (Nix & Busby, 1986; Beaumont, Hughes & Poulsen, 2005; Booth et al., 2014), a climatic envelope method; two regression-based techniques, GAM and GLM (Guisan, Edwards & Hastie, 2002); and four machine-learning algorithms: BRT (Elith, Leathwick & Hastie, 2008), Maxent (Elith et al., 2006; Phillips, Anderson & Schapire, 2006; Merow, Smith & Silander, 2013), RF (Breiman, 2001; Prasad, Iverson & Liaw, 2006), and SVM (Drake, Randin & Guisan, 2006). Detailed information about the functioning of each algorithm can be found in the references cited.

For implementation, we created models using the following R packages: DISMO for Bioclim (Hijmans et al., 2017), SDM for GLM, GAM, BRT, RF, and SVM (Naimi & Araújo, 2016). For GLM, we used a binomial response with a logit link, a quadratic function, and the Akaike Information Criterion (AIC) for stepwise selection (Guisan, Edwards & Hastie, 2002). We implemented GAM with a binomial response and a logit link function (Guisan, Edwards & Hastie, 2002). BRT was parameterized with a learning rate of 0.005, a tree complexity of 5, and a bag fraction of 0.5 (Elith, Leathwick & Hastie, 2008). RF was calibrated with 500 trees (Prasad, Iverson & Liaw, 2006). We used ENMeval (Kass et al., 2021) for Maxent models, testing four regularization multiplier values (0.5, 1, 1.5, 2) and combinations of five feature classes (linear, quadratic, product, threshold, and hinge), with clamping and extrapolation options disabled. The best model for each species was selected using the Akaike Information Criterion corrected for small sample sizes (AICc) (Warren et al., 2014). All output maps were expressed on a continuous scale from 0 to 1.

We allocated 70% of the occurrence records for model calibration and the remaining 30% for validation, applying the same proportions for pseudoabsences where required. The resulting models were then converted into binary maps (presence-absence) using a ten-percentile threshold to minimize overprediction from potentially erroneous data (Radosavljevic & Anderson, 2014). We evaluated the models within time periods using binomial tests that compared the results against random expectations (Anderson, Lew & Peterson, 2003).

Model transferences

The logical procedure to evaluate the capacity of algorithms to transfer niche models across different temporal scenarios involves calibrating a niche model in period 1 and transferring it onto the climatic scenario of period 2, then validating the transference with occurrences from period 2, or calibrating a niche model in period 2 and comparing the two maps (Hijmans & Graham, 2006). However, disparities in the number or environmental distribution of occurrences between periods can affect observed differences between the resulting maps, making it difficult to attribute such differences to algorithm performance alone. To address this issue, we implemented a cross-temporal approach to identify species with similar occurrence data structures in both time periods.

First, we calibrated a model using occurrences and climatic layers from period 1 (“auto1”); then, we generated a second model using the climatic layers from period 1 with occurrences from period 2 (“cross1”). Next, we calibrated a model using occurrences from period 2 with the climatic surfaces from that period (“auto2”) and another model using occurrences from period 1 with the climatic surfaces from period 2 (“cross2”). All resulting maps were converted into binary format, and we compared “auto1” with “cross1” and “auto2” with “cross2” geographically (see next section). If the overall similarity between the two pairs of maps was less than 70%, it indicated significant differences in the number or distribution of occurrences between the time periods, which could hinder the ability of algorithms to transfer models across temporal scenarios. We repeated this procedure for all species and algorithms. Species for which the similarity value was below 70% were excluded from further analyses. Species with suitable datasets for analysis were classified as “control” species, while those without were labeled “without control.” For control species, we proceeded with transferring models from period 1 to period 2 (forecast) and vice versa (hindcast) and subsequently evaluated model performance (Fig. 1).

Figure 1 Methodological approach to evaluate algorithm performance over time.

(A) Ecological niche model calibrated with occurrence data and environmental variables from the same period. (B) Ecological niche model calibrated with occurrence data from one period and environmental variables from a different period. (C) Model transferred to a different period. (D) Geographic validation by comparing the model transferred from one period against the model calibrated in the other period. The example maps correspond to the cotton rat Sigmodon mascotensis, and the climate data for the analysis were obtained from Cuervo-Robayo et al. (2020). The figure was created with Libreoffice-Impress 7.1.1.2 (The Document Foundation, 2020).

Evaluation of algorithm performance

For each algorithm, models transferred from period 1 to period 2 were compared pixel-by-pixel against models calibrated for period 2, and vice versa. We constructed confusion matrices for these comparisons, using the calibration models as references. In the confusion matrix: “a” represents the number of presence pixels correctly predicted by the transferred model (sensitivity), “b” denotes the number of absence pixels incorrectly classified as presence (commission error or false positives), “c” indicates the number of presence pixels incorrectly classified as absence (omission error or false negatives), and “d” corresponds to the number of absence pixels correctly predicted (specificity). We then calculated the following indices to assess model performance: True Skill Statistics (TSS; Eq. (1)), Overlap Index (OI; Eq. (2)), False Negative Rate (FNR; Eq. (3)), and False Positive Rate (FPR; Eq. (4)) (Fielding & Bell, 1997):

(1) TSS=aa+c+db+d−1

(2) OI=aa+c

(3) FNR=bb+d

(4) FPR=ca+c.

TSS measures the accuracy of predictions by comparing the number of correctly predicted pixels to what would be expected by chance. It ranges from −1 (no better than random) to 1 (perfect discrimination), with values above 0.7 considered reliable (Allouche, Tsoar & Kadmon, 2006). OI measures the proportion of overlap between the maps from different time periods indicating their consistency. FNR represents the omission error and measures the rate of overfitting, with values ranging from 0 to 1, and FPR measures the overestimation, also ranging from 0 to 1 (Rebelo, Tarroso & Jones, 2010).

Statistical analysis

To evaluate differences between forecast and hindcast model transfers, we conducted a Mann-Whitney-Wilcoxon test (Pohlert, 2016). We also used a Kruskal-Wallis test to compare the performance of different algorithms in transferring models. When a significant difference was detected, we used a Nemenyi test for pairwise multiple comparisons of mean ranks among algorithms. These statistical tests were performed using the PMCMR package in R (Pohlert, 2016). Additionally, we explored the relationship between the number of occurrence records and model performance—measured by True Skill Statistics (TSS)—for each algorithm and direction (forecast and hindcast) using Pearson’s correlation analyses. All calculations and statistical analyses were carried out in R 3.5 (R Core Team, 2017).

Results

Occurrence data of species

We compiled occurrence data for 117 Mexican endemic rodent species (Ramírez-Pulido et al., 2014). Of these, only 44 species had sufficient unique records (at least 10 per time period) to generate robust models (Supplemental Material S3). Fourteen species were underrepresented in one of the two time periods, while 59 species lacked the minimum of 10 unique records in both periods. Among the 44 species with sufficient data, Peromyscus melanophrys (n = 504), Peromyscus difficilis (n = 440), Chaetodipus arenarius (n = 248), and Sigmodon mascotensis (n = 191) had the highest number of records.

Temporal consistency of occurrence data

We evaluated the temporal consistency of occurrence data for the 44 species with sufficient records using a cross-validation test. None of the algorithms demonstrated high data consistency (defined as >70% similarity) across all 44 species. SVM and RF achieved the highest levels of consistent transferences, successfully applying to 42 species in both hindcasting and forecasting scenarios. In contrast, GLM showed the lowest consistency, with consistent models for 35 species in hindcasting and 36 in forecasting (Table 1).

Table 1 Species with sufficient occurrences.

Algorithm	Hindcasting	Forecasting	
Bioclim	40	38	
Boosted regression trees (BRT)	39	39	
Generalized additive models (GAM)	41	41	
Generalized linear models (GLM)	35	36	
Maxent	38	38	
Random forest (RF)	42	42	
Support vector machine (SVM)	42	42	
Note:

Number of species with consistent occurrence data between time periods for each modeling algorithm.

Niche models and model transferences

Binomial tests revealed that models calibrated with occurrences and climatic layers from the same period (auto1 and auto2) significantly deviated from random expectations for most species, indicating reliable model accuracy. Exceptions included GLM models for Callospermophilus madrensis, Dasyprocta mexicana, and Dipodomys phillipsii in T1 (1950–1979), and Dipodomys phillipsii, Neotamias durangae, Neotoma goldmani, and Oryzomys guerrerensis in T2 (1980–2009); a BRT model for Pappogeomys bulleri in T2, and a Maxent model for Dasyprocta mexicana in T2 (Supplemental Material, Table S4).

We found significant differences between algorithms for transferring models from T1 to T2 and vice versa (H = 78.75 and H = 79.08, respectively, both p < 0.01). RF consistently showed the highest mean TSS ( x¯ = 0.83 ± 0.08) in forecasting, while Maxent in hindcasting ( x¯ = 0.82 ± 0.10). Notably, Maxent showed significantly higher TSS values for hindcasting than forecasting (w = 435, p = 0.002). Conversely, BRT, RF, and Bioclim showed no significant directional differences. Bioclim recorded the lowest in both forecasting ( x¯ = 0.50 ± 0.20) and hindcasting ( x¯ = 0.52 ± 0.21).

Significant differences between algorithms were also evident in OI (H = 27.27, p = 0.014 in hinadcasting and H = 15.93, p < 0.01 in forecasting), with Bioclim ( x¯ = 0.52 ± 0.21) and GAM ( x¯ = 0.65 ± 0.18) showing the greatest variation in forecasting, and Maxent also displaying significant directional differences (w = 382, p < 0.01). RF maintained consistent performance across both directions (forecasting: x¯ = 0.83 ± 0.08; hindcasting: x¯ = 0.80 ± 0.09) (Fig. 2; Supplemental Material, Table S5).

Figure 2 Algorithm performance measured with different metrics.

Performance of niche modeling algorithms for hindcasting and forecasting using different metrics: TSS, true skill statistics; OI, overlap index; FPR, false positive rate; and FNR, false negative rate. Solid red and blue dots represent the median and the upper and lower bars, the interquartile range, and the width of each plot represents the density of observations.

The FPR varied significantly among algorithms in both forecasting and hindcasting (H = 92.237 and H = 77.102, both p < 0.01), with Bioclim showing the highest rates both in forecasting ( x¯ = 0.47 ± 0.22) and hindcasting ( x¯ = 0.47 ± 0.22), and RF the lowest (forecasting: x¯ = 0.09 ± 0.09; hindcasting: x¯ = 0.10 ± 0.08). Maxent showed significant differences in FPR between directions (w = 1062, p = 0.003). The FNR also showed significant differences between algorithms for hindcasting (H = 51.171, p < 0.01) and forecasting (H = 65.92, p < 0.01), where GAM recorded the highest values (Fig. 2; Supplemental Material, Table S5).

Correlation analyses

Our correlation analyses between the number of occurrence records and TSS scores showed weak and non-significant positive relationships for most algorithms (R2 < 0.1, p > 0.05). Exceptions were Bioclim, GAM, and GLM in hindcasting, with GLM also showing a marginal significance in forecasting (Fig. 3). In general, species with larger sample sizes achieved higher TSS scores.

Figure 3 Pearson correlation analyses between the number of occurrence records of each species and model performance measured by true skill statistics (TSS) for each algorithm and direction.

Dots represent individual species and lines the linear trend, red indicates forecasting and blue hindcasting.

Discussion

Ecological niche model transfers are the cornerstone for analyzing species’ distributional impacts due to climate change, yet their reliability is often overlooked and remains untested in most cases. The seven algorithms tested in this study exhibited varying degrees of robustness in their ability to accurately transfer niche models across time, introducing additional challenges in our efforts to anticipate or reconstruct the geographic consequences of climate change. In particular, RF, Maxent, BRT, and SVM consistently performed well, while Bioclim showed the poorest performance. Notably, Maxent exhibited significant sensitivity to the direction of transfer, with marked differences between hindcasting and forecasting. Our results are consistent with previous studies that have observed variations in algorithm performance when transferring niche models to different climatic scenarios (Moreno-Amat et al., 2015; Beaumont et al., 2016; Bell & Schlaepfer, 2016; García-Callejas & Araújo, 2016; Liang et al., 2018; Qiao et al., 2019; Heikkinen, Marmion & Luoto, 2012). However, to our knowledge, no prior studies have examined how occurrence data quality affects algorithm performance in both forecasting and hindcasting.

The presence of non-analog climates is known to affect model transferability between time periods (Sequeira et al., 2018; Charney et al., 2021; Essl et al., 2023). Climatic combinations outside the calibration scenario challenge all algorithms, particularly those with limited extrapolation capacity, such as Bioclim (Qiao et al., 2019). To assess the influence of non-analog climates on algorithm performance, we conducted a Mobility-Oriented Parity (MOP) analysis (Owens et al., 2013), which quantifies the multidimensional similarity between two climatic scenarios (calibration and transfer) and maps areas requiring strict extrapolation, with the smop package (Osorio-Olvera & Contreras-Díaz, 2024) in R. Our results indicate that the areas with dissimilar climatic combinations—and thus where strict extrapolation is needed—are limited across Mexico: from T1 to T2, they occupy 0.58% of the country, and from T2 to T1, 1.7% (Supplemental Material, Fig. S1). Therefore, non-analog climates do not explain most of the observed variation in algorithm performance.

Another crucial factor affecting model performance is data quality, specifically the representation of the environmental combinations that define a species’ ecological niche (van Proosdij et al., 2016; Jiménez-Valverde, 2020). Low-quality data may result from insufficient sampling or environmental bias (Wang & Jackson, 2023). We initially hypothesized that species with fewer occurrences would show poor transferability across all algorithms, as small sample sizes often lead to an incomplete ecological niche characterization. However, our results indicate that sample size impacts algorithms differently: while sample size minimally affected transferability for RF, BRT, and SVM, it significantly impacted Bioclim, GLM, GAM, and, to a lesser extent, Maxent. These findings align with previous studies where Maxent outperformed Bioclim (Hernandez et al., 2006) and GAM (Wang & Jackson, 2023) under small sample sizes.

Sample size also explains performance differences among algorithms in forecasting and hindcasting. In our study, Maxent had the greatest directional difference, showing greater robustness during hindcasting than forecasting (Fig. 2). Notably, Maxent’s sensitivity to sample size was evident only in forecasting (Fig. 3), suggesting that small sample sizes more adversely affect its transferability than other algorithms like RF and BRT. This disparity in performance between Maxent and RF is consistent with Wang & Jackson’s (2023) findings, who recommend RF for small sample sizes.

Modeling algorithm robustness with varying data qualities appears to hinge on their ability to accurately characterize the geometrical complexity of the ecological niche (García-Callejas & Araújo, 2016)—the structural characteristics of the boundary between suitable and unsuitable conditions in environmental space. For species with well-defined niches, like those specialized to specific environments, robust models are generally easier to obtain with less occurrences (Hallman & Robinson, 2020). In contrast, species with broader environmental preferences require more unbiased samples to accurately characterize their more complex niche boundaries. Consequently, some algorithms manage boundary complexity better than others, especially with small sample sizes.

Our research focused on Mexican endemic rodents, most of which are habitat specialists with narrow distributions, although there are some exceptions, such as Peromyscus difficilis (Ceballos & Oliva, 2005). Therefore, we expected that small sample sizes would be sufficient to produce reliable models for most algorithms. However, our findings showed this was not the case, indicating that insufficient sample sizes negatively impacted model robustness and, consequently, temporal transferability. This effect is likely to be even more pronounced for species with broad ecological requirements, regardless of the geographic location or taxonomic group (Moudrý et al., 2024).

We found that Bioclim had the poorest performance in transferring niche models across temporal climatic scenarios. Bioclim, a simple environmental envelope model based on the range of values from occurrence records in the predictor variables (Nix & Busby, 1986), is highly sensitive to extreme values and the number of predictors (Beaumont, Hughes & Poulsen, 2005). Additionally, Bioclim’s quadrangular representation of the ecological niche in environmental space limits its ability to model complex niche geometries. A similar issue may arise with GLM, particularly with small sample sizes (Guisan, Edwards & Hastie, 2002).

In contrast, RF and Maxent exhibited the greatest transfer capacity, followed closely by BRT and SVM. RF also had the most consistent performance between transfer directions. This algorithm has proven robust for transferring models with both virtual (García-Callejas & Araújo, 2016) and real (Mi et al., 2017) species, with good interpolation performance (Bell & Schlaepfer, 2016; Liang et al., 2018), and relatively little overprediction (Mi et al., 2017). However, this robustness comes at the cost of overfitting, which sometimes limits its extrapolation capacity beyond the calibration range (Heikkinen, Marmion & Luoto, 2012). Maxent, on the other hand, is less prone to overfitting, particularly when parameterized ad hoc for specific species (Merow, Smith & Silander, 2013). In summary, the four machine-learning algorithms generally outperformed the two regression-based and climatic envelope algorithms. However, as highlighted in numerous comparative studies, there is no "silver bullet" algorithm that consistently performs best across all data structures (Qiao, Soberón & Peterson, 2015). This is even more evident in model transferability, where algorithm weaknesses are amplified (Pearson, 2006; Moreno-Amat et al., 2015).

A final note of caution is that while our analyses were designed to evaluate algorithm performance for temporal transferability using Mexican endemic rodents as the model system, a potential source of error lies in the continuously updated taxonomy of these species. Our analyses were based on the latest revision of Mexican mammals (Ramírez-Pulido et al., 2014); however, recent proposals suggest species separation for Peromyscus melanophrys, P. furvus, P. levipes, P. zarhynchus, and Osgoogomys banderanus (Lorenzo et al., 2016; Almendra et al., 2018; Cruz-Gómez et al., 2021; Bradley et al., 2022). These taxonomic changes could affect the quantity and spatial structure of occurrences for these species, potentially impacting model performance (Soley-Guardia, Alvarado-Serrano & Anderson, 2024).

Conclusions

Ecological niche modeling is often used to transfer models across temporal scenarios for climate change analysis. However, the suitability of the species for such transfers and the robustness of the chosen algorithms are often overlooked. Our results highlights that the performance of these algorithms, and consequently the reliability of temporal transfers, is primarily influenced by the quality of data. Sample size is a key element for the effectiveness of model transfers since small sample sizes reduce the capacity of some algorithms more than others. Indeed, an algorithm may yield different results for the same species when transferring models to past vs future scenarios, indicating that not all species and algorithms are equally suited for transferring models across temporal scenarios. Among the algorithms evaluated, those capable of modeling complex ecological boundaries with minimal overfitting—such as random forest, Maxent, and boosted regression trees—consistently outperformed the simpler algorithms Bioclim and GLM. Consequently, we strongly recommend a careful assessment of both species and algorithms before proceeding with temporal transfers. In this regard, the retrospective cross-temporal approach presented here offers a valuable alternative.

Supplemental Information

Supplemental Information 1 Supplementary material.

Supplemental Information 2 Transference results.

Each raw indicates the performance result of the model transference (forecast or hindcast) for each species.

Supplemental Information 3 Small mammal species occurrences to build niche models.

Georeferenced occurrences of Mexican endemic small mammals used to build niche modes

Supplemental Information 4 Model validation.

Raws indicate the validation metrics of each species’ model.

Supplemental Information 5 Spatial comparison between time periods to estimate performance metrics.

Supplemental Information 6 Algorithm comparison.

R script to evaluate the environmental and spatial comparison between time periods.

This work is fulfillment of CNMA’s Graduate Doctoral Degree program in Biological Sciences at the Universidad Nacional Autónoma de México (UNAM). CNMA received logistical support from the Posgrado en Ciencias Biológicas and Instituto de Biología-UNAM. We express our gratitude to Livia León-Paniagua and Fausto Mendez for their guidance during the study, to Luis Osorio-Olvera for his assistance with the MOP analysis, Town Peterson for his review to an earlier version of the manuscript, and James Roper, Iván Ray-Rodríguez and two anonymous reviewers for their valuable comments on the manuscript.

Additional Information and Declarations

Competing Interests

Author Contributions

Animal Ethics

Data Availability

The authors declare that they have no competing interests.

Claudia N. Moreno-Arzate conceived and designed the experiments, performed the experiments, analyzed the data, prepared figures and/or tables, authored or reviewed drafts of the article, and approved the final draft.

Enrique Martínez-Meyer conceived and designed the experiments, analyzed the data, prepared figures and/or tables, authored or reviewed drafts of the article, and approved the final draft.

The following information was supplied relating to ethical approvals (i.e., approving body and any reference numbers):

We only used occurrence records of mammal species, which did not involve any management of individuals, so no approval from an Animal Care Committee is required.

The following information was supplied regarding data availability:

The data is available in the Supplemental Files.

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
