# Peer review of "A retrospective approach for evaluating ecological niche modeling transferability over time: the case of Mexican endemic rodents"

_PeerJ, doi:10.7717/peerj.18414_

## Round 0.1 · original submission · Major Revisions

Please read the reviewer's comments carefully. I am also sending an annotated copy of your manuscript with my observations. Rather than repeat anything the reviewers said, I'll point out a few comments from my perspective.

I only commented at length in your introduction, and many of those comments are about the writing style and English. So please note that those kinds of comments are meant to be helpful and constructive for writing more succinctly and clearly in English. Other comments are more substantive and have to do with the logic and content. I hope you will pay attention to those in the context of those of the reviewers. Remember, in the reviews, you do not have to agree or change everything the reviewers ask for, IF you can explain. But, recognize that a reviewer may misunderstand because of style and not ideas, so consider rewording to clarify as necessary.

Please pay particular attention to the Discussion. You started weakly by not emphasizing your results and ended weakly by not concluding which models are best (even if some are essentially equal, they might have methodological differences that make one more practical than another, such as computing power needed).

·

Basic reporting

This manuscript, titled 'A retrospective approach for evaluating ecological niche modeling transferences over time: The case of Mexican endemic rodents,' evaluated the performance and consistency of seven popular algorithms for transferring niche models to past (hindcast) and future (forecast) climatic conditions for Mexican endemic rodents as the study model.

This work represents a significant advance and is worthy of publication in this prestigious journal, albeit with some changes. I have attached the manuscript with comments, particularly focusing on the results and discussion.

I strongly believe that this work is of great interest and that the findings are highly important. Therefore, the publication of this manuscript will be of general interest and will have a significant impact.

The interpretations are well-supported by rigorous and meticulous results. The figures are appropriately chosen and presented.

Experimental design

It is well-structured and robust. However, it should be implemented (I attached the manuscript with the comments).

The R codes should be translated.

Validity of the findings

This work represents a significant advance as they compare seven algorithms

Additional comments

Please consider the supplementary material, is really good, and should be incoporated to the text. When referring to specific tables, please indicate which table you are referencing.

Reviewer 2 ·

Basic reporting

Dear Editor,

The article brings very interesting ideas for testing niche modeling algorithms and has very important contribution to the field. The authors showed that it is important to test the algorithms available prior to use it and propose a workflow to test them. I recommend the article publication after some major review.
The main point is that the discussion final two paragraphs and conclusion have some flaws specially on the way author write the text and make it too generalist beyond their study design and results. Another important issue is regarding to how authors set their modeling geographical scope for each species. They consider ecoregions with caring that ecoregions may change from past to future as the species they are modeling. I would recommend to stablish a modeling scope just with a buffer around the occurrence records of each species instead of using ecoregions. A third and final point is that authors take some conclusions from table 1 that are not clear at the table.
Best regards,

Specific comments:

Title suggestion: Evaluating ecological niche models algorithms transferences over time
41: “Years now” sounds a bit strange
51: It is not landscape but environment or geographical space
128: What happens if ecoregions change over the time? Would your modeling scope identify it? My suggestion is to use a buffer of some kilometers around the kwon occurrence records.
242: It is not only maxent. Many other performed well for 44 species.
288 to 294: It should be at the Methods not at Discussion
371 and 319: Where is it really “finally”?
392: You did not test for quality and quantity of data?
395: Species or models algorithms are not suitable?
399 to 401: It seems to be a general statement from literature or AI.

Experimental design

Dear Editor,

The article brings very interesting ideas for testing niche modeling algorithms and has very important contribution to the field. The authors showed that it is important to test the algorithms available prior to use it and propose a workflow to test them. I recommend the article publication after some major review.
The main point is that the discussion final two paragraphs and conclusion have some flaws specially on the way author write the text and make it too generalist beyond their study design and results. Another important issue is regarding to how authors set their modeling geographical scope for each species. They consider ecoregions with caring that ecoregions may change from past to future as the species they are modeling. I would recommend to stablish a modeling scope just with a buffer around the occurrence records of each species instead of using ecoregions. A third and final point is that authors take some conclusions from table 1 that are not clear at the table.
Best regards,

Specific comments:

Title suggestion: Evaluating ecological niche models algorithms transferences over time
41: “Years now” sounds a bit strange
51: It is not landscape but environment or geographical space
128: What happens if ecoregions change over the time? Would your modeling scope identify it? My suggestion is to use a buffer of some kilometers around the kwon occurrence records.
242: It is not only maxent. Many other performed well for 44 species.
288 to 294: It should be at the Methods not at Discussion
371 and 319: Where is it really “finally”?
392: You did not test for quality and quantity of data?
395: Species or models algorithms are not suitable?
399 to 401: It seems to be a general statement from literature or AI.

Validity of the findings

Dear Editor,

The article brings very interesting ideas for testing niche modeling algorithms and has very important contribution to the field. The authors showed that it is important to test the algorithms available prior to use it and propose a workflow to test them. I recommend the article publication after some major review.
The main point is that the discussion final two paragraphs and conclusion have some flaws specially on the way author write the text and make it too generalist beyond their study design and results. Another important issue is regarding to how authors set their modeling geographical scope for each species. They consider ecoregions with caring that ecoregions may change from past to future as the species they are modeling. I would recommend to stablish a modeling scope just with a buffer around the occurrence records of each species instead of using ecoregions. A third and final point is that authors take some conclusions from table 1 that are not clear at the table.
Best regards,

Specific comments:

Title suggestion: Evaluating ecological niche models algorithms transferences over time
41: “Years now” sounds a bit strange
51: It is not landscape but environment or geographical space
128: What happens if ecoregions change over the time? Would your modeling scope identify it? My suggestion is to use a buffer of some kilometers around the kwon occurrence records.
242: It is not only maxent. Many other performed well for 44 species.
288 to 294: It should be at the Methods not at Discussion
371 and 319: Where is it really “finally”?
392: You did not test for quality and quantity of data?
395: Species or models algorithms are not suitable?
399 to 401: It seems to be a general statement from literature or AI.

Additional comments

Dear Editor,

The article brings very interesting ideas for testing niche modeling algorithms and has very important contribution to the field. The authors showed that it is important to test the algorithms available prior to use it and propose a workflow to test them. I recommend the article publication after some major review.
The main point is that the discussion final two paragraphs and conclusion have some flaws specially on the way author write the text and make it too generalist beyond their study design and results. Another important issue is regarding to how authors set their modeling geographical scope for each species. They consider ecoregions with caring that ecoregions may change from past to future as the species they are modeling. I would recommend to stablish a modeling scope just with a buffer around the occurrence records of each species instead of using ecoregions. A third and final point is that authors take some conclusions from table 1 that are not clear at the table.
Best regards,

Specific comments:

Title suggestion: Evaluating ecological niche models algorithms transferences over time
41: “Years now” sounds a bit strange
51: It is not landscape but environment or geographical space
128: What happens if ecoregions change over the time? Would your modeling scope identify it? My suggestion is to use a buffer of some kilometers around the kwon occurrence records.
242: It is not only maxent. Many other performed well for 44 species.
288 to 294: It should be at the Methods not at Discussion
371 and 319: Where is it really “finally”?
392: You did not test for quality and quantity of data?
395: Species or models algorithms are not suitable?
399 to 401: It seems to be a general statement from literature or AI.

Reviewer 3 ·

Basic reporting

It seems to be a well-made, transparent, and sufficiently explained manuscript.
On the other hand, it is not enough for the raw data to have only the species name, latitude, and longitude. I think it is convenient that they include the information on the voucher or identified citizen science specimen. The authors mention that the taxonomy has changed, so I consider it relevant to have that information to verify the taxonomic identification for those interested in this aspect of the study (e.g., mammalogists, biogeographers).

Experimental design

In the abstract and introduction, the authors mention that they are interested in the performance of seven different and very popular algorithms for model transfers using a case study (endemic rodents from Mexico). Based on the arguments offered in the introduction about the -general- way of modeling the species niche by each algorithm, is it possible to have some predictions or anticipate some results? For example, say that given XXX characteristics, presence-only algorithms will show better values of XXX than XXX algorithms. If that suggestion is valid, it would be helpful to have a richer and more interesting discussion about why some algorithms perform better than others in cross-time transferences.

Validity of the findings

The case of non-analogous climates between periods is mentioned in the introduction and part of the discussion. However, it is something that the authors do not explore in depth in their study. First, are there similar climates between the two study periods (1950-1979 and 1980-2009)? I can bet not, but it should be explicitly clarified. Some algorithms could perform well between different times if non-analogous climates are not an issue, but their performance could worsen due to the need to extrapolate if non-analogous climates become common. This point is relevant to readers interested in transfers to more distant times (e.g., the late Pleistocene and the end of the current century).

The Discussion about why the algorithms perform differently when used for cross-time transfers can be enriched. As presented now, the Discussion seems more like a summary of results and whether it corroborates what other authors said. However, it can be enriched if the reasons for such differences are further explored. In this sense, most of the literature cited in the Discussion (although very relevant) is from more than 5 years ago. I know we should not detract from scientific literature just because it was published several years ago. However, I wonder why there is no reference to more current literature (especially in a field with so many publications generated almost daily). Some relevant studies I recommend are:
https://doi.org/10.1016/j.ecolmodel.2021.109671
https://doi.org/10.1111/2041-210X.12397
https://doi.org/10.1016/j.ecolmodel.2021.109502
https://doi.org/10.1111/ecog.06852

Near the end of the Discussion, the authors call out the limitations of using a relatively outdated taxonomy for their study. The authors' attention to this fact is acknowledged, but the message is unclear. I consider this not a limitation in the strict sense of what they want to evaluate (performance between algorithms for niche modeling transferences), but rather for those interested in the distribution of endemic rodents in Mexico. This critical issue must be made known to users who wish to interpret or use these potential distributions for other purposes (e.g., biogeography, conservation, etc.). In this sense, I think the authors should paraphrase this paragraph from their Discussion.

In Conclusions, the authors say "not all species are suitable candidates for transferring models to alternative climatic scenarios; only those species with sufficient occurrence records to produce robust niche models, should be considered for projections time periods". We all agree and this is not a new finding. I recommend eliminating this phrase or paraphrasing it so that it highlights something new about this study.

Additional comments

Please carefully review the writing of scientific names throughout the manuscript.

---

## Round 0.2 · Minor Revisions

Please check the annotated manuscripts for a few minor suggestions, and the reviewers for their several suggestions to improve the manuscript.

·

Basic reporting

This manuscript "A retrospective approach for evaluating ecological niche modeling transferability over time: the case of Mexican endemic rodents", after the revision, the manuscript has improved a lot, and all my suggestions have been implemented.
There are some minor erros like in like 92 in the sentence: "Rubidge et al., 2011; Moreno-Amat et al., 2015; García-Callejas & Araújo, 2016) These studies", you need to put the point.
So, please, check the grammar for possible errors.

Experimental design

It is well-structured and robust.

Validity of the findings

I firmly believe that this work is highly significant and the findings are crucial, such a studyy case. Consequently, publishing this manuscript will garner broad interest and make a substantial impact. The interpretations are well-supported by the results, which have been obtained with rigorous and meticulous methods. The figures are both appropriately selected and effectively presented

Reviewer 2 ·

Basic reporting

The authors did a significant rewrite of the manuscript.

After carefully reviewing the manuscript and considering the specific comments I had previously made, I can now recommend the article for publication in PeerJ.

Experimental design

not applicable

Validity of the findings

not applicable

Additional comments

not applicable

Reviewer 3 ·

Basic reporting

The English has been improved for this revised version, and references have been made to more current publications on the subject.

Experimental design

This revised version already included the necessary analyses given the original research questions.

Validity of the findings

Overall I think they are appropriate, but I have made a couple of suggestions to try to improve them.

Additional comments

The authors have made a great effort to enrich their research and the manuscript, which is praiseworthy and appreciated. Rest assured, the following two minor recommendations are meant to further enhance the quality of your manuscript.

COMMENT 1
One of the authors' conclusions is still (copied from the Abstract): "These findings underscore that not all species or algorithms are suited for temporal projections. Therefore, we recommend testing multiple algorithms using a retrospective approach before applying models to future scenarios". For me, it is still a bit confusing since the research question has to do with the performance of the algorithms -using a taxonomic group in a very particular study region as an example. I think that the conclusion that the authors offer still seems to me to be inappropriate for their initial research question. I hope not to confuse the authors further, but I think their conclusion should refer to the particularities of the algorithms and not to the study case taxon. In the conclusion, a sentence seems more appropriate to me: "we strongly recommend a careful assessment of both species and algorithms before proceeding with temporal transfers." However, I think the authors could enrich it by mentioning what people should look at to select the best combination to have more "realistic" estimates.

COMMENT 2
Another point that could be strengthened in the discussion is the particularities of the study area that the authors use as a case study. If people work in other regions of the planet with different characteristics (e.g., non-tropical or subtropical region, desert region, high mountain region), could similar results or behaviors be expected from the algorithms? Why?

---

## Round 0.3 · accepted · Accept

I still have one suggestion I think you should consider. Your figure 3 has a couple of issues. The x-axis says "Occurrence data" but doesn't say what the numbers represent? Also, the distribution of that variable clearly appears to be log normal. I would think you could make it much more informative by clearly stating what the numbers on the line mean (number of occurrence records, not Occurrence data, although occurrence counts might be more reasonable, regardless, it should be clearer) and using a log SCALE (not log transformed values). In R, if you're using ggplot2, that would require a line that says "scale_x_log10() +" It will not change anything else about your results, and will only make that figure more informative.

We appreciate the effort you've taken to improve the quality of this manuscript.